# Surgery Versus Stereotactic Radiotherapy in Patients over 75 Years Treated for Stage IA–IIA NSCLC

**DOI:** 10.3390/cancers17040677

**Published:** 2025-02-17

**Authors:** Galdric Oliver, Mohamed Boucekine, Anne-Laure Couderc, Alex Fourdrain, Audrey Zaccariotto, Isabelle Pougnet, Bertrand Kaeppelin, Pascal Alexandre Thomas, Laetitia Padovani

**Affiliations:** 1Oncology Radiotherapy Department, CRCM Inserm, Genome Instability and Carcinogenesis, Assistance Publique des Hôpitaux de Marseille, Aix Marseille University, 13284 Marseille, France; audrey.zaccariotto@ap-hm.fr (A.Z.); isabelle.pougnet@ap-hm.fr (I.P.); bertrand.kaeppelin@ap-hm.fr (B.K.); laetitia.padovani@ap-hm.fr (L.P.); 2Center for Studies and Research on Health Services and Quality of Life, Aix Marseille University, 13284 Marseille, France; mohamed.boucekine@ap-hm.fr; 3Internal Medicine, Geriatric and Therapeutic Unit, University Hospital (AP-HM), 270 Boulevard de Sainte Marguerite, 13009 Marseille, France; anne-laure.couderc@ap-hm.fr; 4CNRS, EFS, ADES, Aix-Marseille University, 13284 Marseille, France; 5Department of Thoracic Surgery, Diseases of the Esophagus & Lung Transplantation, Hôpital Nord & CRCM, Inserm UMR 1068, CNRS, UMR 7258, Assistance Publique-Hôpitaux de Marseille, Aix-Marseille University, 13284 Marseille, France; alex.fourdrain@ap-hm.fr (A.F.); pascalalexandre.thomas@ap-hm.fr (P.A.T.)

**Keywords:** elderly patients, stereotactic radiotherapy, localized lung cancer, real-world evidence, lung surgery, propensity score matching, tree-based model, French

## Abstract

Lobectomy with lymph node dissection remains the gold standard for early-stage Non-Small Cell Lung Cancer (NSCLC) but carries a higher early mortality risk, particularly in elderly patients with comorbidities. Stereotactic radiotherapy (SABR) has shown promising local control and tolerance in this population. This retrospective study (2012–2022) compared 127 surgical patients and 85 SABR patients using propensity score matching. Five-year survival was higher in the surgery group (47.3%) than in the SABR group (31.5%), though the difference was not statistically significant (*p* = 0.068). Factors influencing early mortality included gender, WHO status, FEV1, and treatment type. A novel decision-making tool is proposed to optimize patient selection.

## 1. Introduction

Lung cancer incidence increases proportionately with age [1]. The early stage represents around 25% of non-small cell lung tumor (NSCLC) at diagnosis [2]. Complete surgical excision with mediastino-hilar lymph node dissection in operable patients with localized NSCLC is still the standard of treatment (stage IA–IIIA), achieving local control of around 90%, with an early mortality rate of around 5% within 90 days [3,4,5]. However stereotactic ablative radiotherapy (SABR) provides local control approaching that of surgery, with an excellent safety profile [6,7]. The various attempts at randomized trials to compare these two types of management were closed early due to low recruitment [8,9,10]. Several studies and meta-analyses, including propensity score-matching analysis, have found no significant differences in overall and disease-free survival between SBRT and surgery after adjustment [11,12,13]. Statistical methods, such as propensity score analysis, are essential in non-randomized studies due to the significant bias in survival outcomes among patients treated with radiotherapy who were deemed ineligible for surgery because of comorbidities that limit their life expectancy [14]. In this context, SABR has been admitted as standard treatment for patients not eligible for surgery and remains an alternative for operable patient in whom other causes of mortality due to comorbidities compete with those related to cancer [15]. A major question remains the place of SABR in elderly patients. The SABR study for the management of early-stage NSCLC showed good results in terms of safety and survival for this population [16,17,18,19,20]. The most appropriate therapeutic strategy for these patients is still the subject of much debate in the literature [21].

In this context, we carried out a large single-center retrospective study to compare surgery and SABR for the curative management of patients ≥ 75 years of age treated in routine for stage IA–IIA NSCLC.

## 2. Materials and Method

### 2.1. Population

This monocentric retrospective study included patients ≥ 75 years of age who received surgery or SABR for stage IA–IIA NSCLC (tumors less than 5 cm in size with no lymph node spread (cN0) or distant spread (cM0)). Previous treatment for another cancer was not an exclusion criterion. The inclusion period was from 1 January 2012 to 31 December 2022. Data extraction was performed on 15 June 2023. Surgical patients were identified from a prospectively maintained lung cancer database contributing to EPITHOR^®^, the French Society of Thoracic and Cardiovascular Surgery general thoracic surgery registry. Patients’ consent was obtained for entry into the database, and they were aware that this information would be used anonymously for research purposes. The characteristics of patients treated with SABR for the same pathology were collected retrospectively from the patient records of the same institution. Patients were informed orally of the possibility of their data being used for research purposes, and they signed consent documents.

According to French guidelines, the standard pre-therapeutic work-up included a thoracic and abdominal Computed Tomography scan (CT scan), metabolic imaging using integrated Positron Emission Tomography–Computed Tomography (PET-CT) and magnetic resonance brain imaging [22]. Operable patients with an N2 ≥ 1 cm status on a CT or PET scan underwent mediastinal staging by endobronchial ultrasound (EBUS) or mediastinoscopy [23]. Only N0 patients post-mediastinoscopy and N0 patients on a CT and PET scan were included. All patients in the EPITHOR database had histological confirmation of NSCLC.

Lung biopsies were taken whenever possible in patients for whom stereotactic radiotherapy was indicated. The others were treated without anatomopathological confirmation, based on iconographic and clinical evidence, following a decision by a multidisciplinary consultation meeting.

### 2.2. Geriatric Score

A score of cumulative comorbidities common to patients treated by SABR or surgery was calculated, as previously published by our team [24]. Each comorbidity received 0, 1, or 2 points depending on whether it was controlled by treatment, uncontrolled or untreated. An additional 3 points were awarded for each comorbidity that had an impact on the proposed treatment. This score is relevant because it specifically considers pathologies that are at risk of complications with the proposed treatment. It also considers the stability of the comorbidity and whether it is being treated. For each comorbidity, the count can range from 0 to 5.

For example, in the case of a patient with COPD well controlled by medication and a history of coronary artery disease treated with stenting, without ischemic manifestations and on antiplatelet therapy, the index is calculated as follows:Tobacco: weaned, hence no direct impact, but it interferes with management due to its association with cancer (0 + 3 = 3);COPD: well controlled (0) but interferes with pulmonary function during treatment (3);Coronary artery disease: stented (0) but interferes with cardiovascular risk considerations when selecting cancer treatment (3);Antiplatelet therapy: no direct impact (0), but it interferes with surgery (3), though not with radiotherapy (0).

Total index for the patient: 12 (if surgery is planned) or 9 (if radiotherapy is chosen). There may be inter-individual or intra-individual variations, but overall, the index provides valuable and informative insight.

Patients’ general conditions were assessed using the WHO performance index and nutritional status was assessed using body mass index prior to treatment. Respiratory function was assessed by measuring forced expiratory volume in one second (FEV1) and carbon monoxide diffusion capacity (DLCO).

### 2.3. Radiotherapy Treatment

Radiotherapy techniques were modified during the inclusion period: between 2012 and 2019, patients were treated with TomoTherapy^®^. During this period, the dose was prescribed at 100% isodose. The most used fractionation was 60 Gy in 5 fractions based on the dose constraints of Timmerman et al. [25].

From 2020 to 2022, patients were treated by Cyberknife^®^ (Accuray, Sunnyvale, CA, USA), or by a VERSA HD^®^ (Elekta, Stockholm, Sweden) dedicated particle accelerator. The dose was prescribed at 60% to 80% isodose, depending on the device used. The implementation of type C algorithms resulted in the prescribed dose being modified to 55 Gy in 5 fractions. The dose constraints were in line with recommendations from various publications [26,27,28].

The dose and the fraction number were adapted in the case of a central tumor or according to the patient’s history, particularly in the case of re-irradiation or severe cardiopulmonary comorbidities. The organs at risk considered were the bronchi, trachea, medullary canal, esophagus, heart and large vessels, lungs, skin, and ribs in the case of peripheral tumors and the brachial plexus in the case of apical tumors.

### 2.4. Surgery

The surgical approaches were a thoracotomy (open), a multiport video-assisted thoracoscopy (VATS), or 3- or 4-arm robot-assisted thoracoscopy (RATS). The standard lung resection was a lobectomy. Sublobar resections were performed in selected high-risk patients only. All patients underwent lymph node dissection. Intraoperative lymph node assessment was described by the surgeon in the operative report as none, sampling, selective (lobe-specific), or systematic lymphadenectomy.

### 2.5. Patients Follow-Up

After surgery, clinical assessment was performed at 1 month and then on a regular basis. Cerebral, thoracic, and abdominopelvic cross-sectional imaging was performed 3 months after treatment, then every 6 months for 2 years and annually thereafter. A PET scan was requested if there was any doubt about a local recurrence on a radiotherapy scar or if a lesion suspected of malignancy appeared. Complications were graded according to CTCAE version 5 for patients treated with radiotherapy and were classified according to the Clavien-Dindo classification for patients treated by surgery.

The clinical status and causes of death, if any, were obtained for all patients from medical records or by contacting the attending physician and family. The exact date of death was obtained from the register of deceased persons maintained and updated monthly by the “Institut national de la statistique et des études économiques”.

Unplanned hospitalizations were collected to indirectly assess patients’ loss of autonomy.

### 2.6. Statistical Analysis

The primary endpoint was overall survival (OS).

Disease-specific criteria of interest included locoregional (EFLRS) and metastatic (EFMS) progression. The definition of locoregional progression included radiotherapy scar recurrence, surgical suture recurrence, recurrence in the form of a new lung nodule in the same lobe, or mediastino-hilar nodal recurrence.

Progression was documented by CT scan, sometimes combined with metabolic imaging and histological documentation.

The criteria of interest for assessing treatment toxicity were early mortality within 90 days and the rate of treatment-related unplanned hospitalization.

Quantitative variables are presented as mean ± standard deviation or median and interquartile range (IRQ). Categorical variables are presented in terms of numbers (%). For quantitative variables, comparisons of means between groups were made using a Student or Mann–Whitney test, depending on the hypotheses applied. For qualitative variables, comparisons were made using the Chi-2 or Fisher exact test, depending on the assumptions made.

To ensure that the surgery and radiotherapy groups were comparable, a propensity score was used to match the following parameters: tumor size, sex, age, comorbidity index, smoking status, and WHO status. Balancing of matching variables can be seen in Figure 1. A ratio of 2 surgical patients to 1 patient treated with radiotherapy was used. For each matching variable, the quality of the match was checked using the standardized mean difference, which had to be less than 0.1. The R software (version 4.3.1) package “matchIt” was used.

Overall survival and progression in the radiotherapy and surgery groups were estimated using Kaplan–Meier curves and the log-rank test. Univariate and multivariate Cox analyses were performed to adjust for other risk factors. Variables with a *p* < 0.2 and not strongly correlated were included in the multivariate. This threshold of 0.2 was chosen according to the number of variables and events in our study. It is a guide to avoid missing any important variable in the adjustment of the multivariate model and to eliminate those that are noise. Analyses were performed using the “survival” package of R software. A *p* < 0.05 was considered significant.

A CART decision tree analysis was performed to try to identify clusters of patients potentially at risk for overall survival and early death. The selected clusters were then compared using Kaplan–Meier curves and the log-rank test.

## 3. Results

### 3.1. Patients Characteristics

A total of 389 elderly patients were treated for stage cIA–IIA lung tumors between January 2012 and December 2022 at the Marseille University Hospital: a total of 288 underwent surgery and 101 received ablative stereotactic radiotherapy. The median follow-up was 30 months for the surgical population (IC95% = [24.9;35.8], min = 0, max = 119) and 23 months for the radiotherapy population (IC95% = [19.4;27.1] min = 3, max = 104). The characteristics of the two populations before matching are shown in Table 1. The mean age on the day of surgery was 79 years. In the radiotherapy group, 18 patients under the age of 75 (ranging from 68 to 74 years) were included in the analysis due to their comorbidity profiles being comparable to the rest of the population. Of all the patients treated with radiotherapy, half did not have a histological confirmation of the diagnosis.

As expected, before matching, the performance status, comorbidity index, and respiratory function criteria were in favor of the surgery group. The values used for FEV1 and DLCO are the gross pre-therapeutic values, not the predictive ones. Of these 389 patients, 212 were matched with a propensity score according to the chosen variables: 85 in the radiotherapy group and 127 in the surgery group. The variables of interest chosen for matching were evenly distributed. The distribution of patient characteristics after matching is shown in Table 2. The mean duration of follow-up in the population of 212 matched patients analyzed was 23.4 months in the surgery group (IC95% = [19;28.7], min = 3.28, max = 104) and 26.6 months in the radiotherapy group (IC95% = [19.4;34.1], min = 0, max = 108).

Technical data on surgery and radiotherapy are shown in Table 3.

### 3.2. Overall Survival

In the general population before matching, the overall survival of patients operated on at 1, 3 and 5 years was 89.6% [86.1–93.4], 77.06% [71.6–82.9], and 61.28% [54.0–69.5] respectively, compared with 89.59% [83.7–95.9], 52.95% [42.7–65.6], and 27.32% [17.1–43.7] in the radiotherapy group, *p* < 0.0001 (Figure 2). There were 90 deaths in the surgery group (31.2%) and 52 in the radiotherapy group (52.5%) over the period studied.

After matching, the overall survival of patients operated on at 1, 3, and 5 years was 83.87% [77.4–90.8], 73.61% [65.2–83.1], and 47.30% [36.1–62.0], respectively, compared with 88.8% [82.2–96.0], 57.1% [46.3–70.4], and 31.5% [19.9–49.9] in the radiotherapy group *p* = 0.068. There is a trend in favor of the surgery group, but the difference remains not significant. In multivariate analysis, only WHO status had a significant impact on overall survival (HR = 2.53 [1.38–4.70] *p* = 0.003) (Table 4). Treatment strategy has not been found as significantly associated with overall survival.

### 3.3. Locoregional Recurrences

In the general population, 27 patients in the surgery group (9.4% [6.4–13.5]) and 23 patients in the radiotherapy group (22.8% [15.2–32.4]) developed locoregional recurrence (*p* = 0.001).

In the general population, the mean time between treatment and local or regional recurrence was 24.6 months [21.5–29.6] in the radiotherapy group and 34.6 months [31.4–37.9] in the surgery group (*p* = 0.03). Most of these diagnoses were made early after treatment, reinforcing the need for close follow-up in the early years.

After matching, 15 patients in the surgery group (11.8% [7.9–21.4]) and 18 patients in the radiotherapy group radiotherapy (21.2% [13.3–31.6]) presented locoregional recurrences.

The probability of survival without locoregional recurrence (EFLRS) for patients treated with surgery at 1 and 3 years was 93.1% [88.4–98.2] and 89.4% [83.3–95.9], respectively, compared with 94.5% [89.4–99.9] and 64.7% [52.1–80.3] for patients treated with radiotherapy (*p* = 0.052). Considering the occurrence of death as a competing risk in the occurrence of the local recurrence event reduced the difference between the two groups: *p* = 0.11; HR 1.73 (0.878–3.41).

Of the 101 SABR, 29 had a recurrence (28.7%). Of these 29 recurrences, only 9 were exactly local in contact with the irradiation field (31%), 14 were nodal or in the form of a pulmonary nodule at a distance from the irradiation field (48.3%), and 6 were metastatic (20.7%). One third of exactly local recurrences were in central tumors and one third had a BED of less than 100 Gy.

The mean time between treatment and recurrence was 15.9 months.

Ten of these recurrences were documented by imaging + biopsy and nineteen by imaging alone. Of the purely local recurrences, three were documented histologically.

Of the 29 patients, 25 were able to receive new oncological treatment for their recurrence, 2 received exclusive palliative care, and the information is unknown for 2 others. A total of 15 of the 25 patients received a new local treatment: 12 re-irradiations (stereotactic or conventional), 1 radiofrequency, 1 cryotherapy, and 1 adrenalectomy in the case of metastatic oligoprogression.

A total of 10 out of 25 patients received systemic therapy after progression: chemotherapy (n = 7), immunotherapy (n = 2), and targeted therapy (n = 1). Of the 29 patients with recurrence, 16 died. The mean time from recurrence to death was 14.6 months, with only one death within 6 months of treatment.

These data show that most patients treated with radiotherapy are retreated for recurrence, and that recurrence does not appear to be the cause of death in the majority of cases.

### 3.4. Metastatic Recurrence

In the general population, 22 patients in the surgery group (7.6%) had a metastatic recurrence, compared with 6 patients in the radiotherapy group (5.9%). This difference was also seen in the matched population, with 11 patients in the surgery group (8.6%) and 5 patients treated with SABR (5.8%) presenting with a metastatic event. This difference may be explained by the fact that there are more deaths in the SABR population and therefore no time to develop metastases.

### 3.5. Tolerance

In the matched population, 10 patients treated with surgery (8.3% [4.0–14.4]) and 1 patient treated with SABR (1.2% [0.06–7.3]) died within 90 days of treatment. This result was non-significantly in favor of radiotherapy (*p* = 0.063).

No unplanned hospitalization related to stereotactic radiotherapy treatment was reported. Eleven patients (8.7%) had to be readmitted for complications related to the surgery. The reasons for hospitalization were mainly related to infectious pathologies.

### 3.6. Sub-Group Analysis

The CART subgroup analysis of overall survival is shown in Figure 3, highlighting the major impact of DLCO and comorbidity index. Overall, three distinct subgroups of patients are clearly identified based on their respiratory function, WHO status, and comorbidities within the matched population.

In the matched population, the four factors affecting early death were sex, WHO status, treatment group, and FEV1 (Figure 4). We can deduct from this that in male patients, with a WHO score of 0 or 1 and FEV1 < 85% of the predicted value, receiving surgery compared with SABR is more likely to result in death within 90 days.

Female gender appears to be a protective factor, with none of the 65 women experiencing the event.

## 4. Discussion

To our knowledge, this is the largest, matched, real-life study comparing SABR with surgery in elderly patients with early-stage NSCLC, and offering a specific therapeutic decision support tool for this population.

We confirm the excellent tolerance and local control of stereotactic radiotherapy (91.09%). As expected, postoperative morbidity and mortality resulted in an excess of early deaths within 90 days in patients who underwent surgery. However, the results seem to confirm the surgical indication as the long-term therapeutic standard in elderly patients: OS at 5 years and EFLRS at 3 years, respectively; 47.30% and 89.4% for operated patients compared with 31.50% and 64.7% for patients treated with SABR. The overall survival of patients treated with surgery or SABR is in line with that reported in the literature [3,4,5,6]. For the matched population, the difference is no longer significant, but there is a trend in favor of surgery.

The difference in metastatic recurrence (8.6% in the surgery group versus 5.8% in the radiotherapy group) may be explained by the fact that there are more deaths in the SABR population and therefore no time to develop metastases. There are more tumors classified as T2 in the surgery group, with a greater risk of metastasis. Twice as many locoregional recurrences occurred in the radiotherapy group. However, a study of these recurrences found few true local failures of SABR besides metachronous lymph node and lung progression (9 purely local recurrences compared with 14 regional recurrences). Furthermore, the SABR technique has improved over the years and in this study most patients treated with radiotherapy received a dose with a BED < 100 Gy due to both the learning curve and the central or ultra-central tumor locations, which have a direct negative influence on local control [29]. In addition, the diagnosis of recurrence after SABR remains very complex and is often debatable [30]. In contrast to the evaluation in the surgical population, local recurrence may have been overestimated in the radiotherapy group.

This result must be interpreted with a few limitations in mind:Despite negative CT and PET scans, 36 of the 270 patients with lymph node dissection (13.3%) had lymph node upstaging and were therefore in more advanced stages. This information was ignored in patients treated with SABR with an increased risk of regional recurrence. Razi et al. demonstrated that the overall survival benefit of surgery compared with radiotherapy in an elderly population is significant only if lymph node surgery is associated. The same study suggests that overall survival is improved in part by adjuvant treatment in patients with tumor stage II or III after surgery [31];Despite statistical balancing on prognostic factors, patients in the radiotherapy group remain more fragile, often pre-treated and with respiratory reserve that does not allow them access to surgery. Their profile means that, apart from purely oncological considerations, they are at greater risk of dying from intercurrent diseases.

Interestingly, the CART analysis of overall survival in the matched population highlights the major impact of respiratory status, WHO status, and comorbidity index (Figure 3). The comorbidity index appears to be the second most discriminating variable after DLCO. This highlights the importance of considering patients’ underlying conditions when estimating their survival and recommending the most appropriate treatment. A comprehensive geriatric assessment, including the use of scores as outlined in this study, seems essential for informed therapeutic decision-making.

The treatment group did not appear to have a discriminatory effect on overall survival in the multivariate and subgroup analyses. This suggests that patients’ survival is mainly affected by their comorbidities rather than their cancer, which may have benefited from second-line treatment in the majority of patients after recurrence.

Few published studies have reported a comparable result. This is probably due to the fact that our study is one of the few to provide rigorous information on comorbidities, general condition, disease characteristics, and treatment techniques.

One of the major findings of our study is the increased risk of early death after surgery, illustrated in Figure 4. The survival curves cross at 15 months, reflecting an initially greater benefit of SABR due to better immediate tolerance. In the long term, surgery offers a higher probability of survival.

It therefore appears that the challenge in this population of elderly subjects is to discriminate between the sub-group of patients at risk of a postoperative complication to offer them SABR and thus improve their survival. In the matched population, clusters were created according to the risk of early death within 90 days.

After subgroup analysis using the CART method, we are proposing a therapeutic decision aid for elderly patients already considered eligible for surgery. This decision aid, which considers gender, WHO status, and FEV1 status, will make it possible to redirect these patients eligible for surgery towards SABR and thus help to improve their overall survival.

Our study has several limitations linked to its retrospective nature, with some data missing, and information on specific cancer deaths and on cancer treatments appearing prior to inclusion in the study. The median follow-up is correct but remains too short for the last patients included.

## 5. Conclusions

In conclusion, our results seem to confirm a trend towards superiority of surgery in terms of overall survival and locoregional control, with nonetheless excellent disease control and tolerability after SABR. Our study highlights the strong impact of early death after surgery. We believe that the future challenge for this population lies in a more relevant therapeutic selection. In this context, we are proposing for the first time a therapeutic decision support tool to improve the selection of patients eligible for surgery and thus further improve the overall survival of this population.

## Figures and Tables

**Figure 1 cancers-17-00677-f001:**
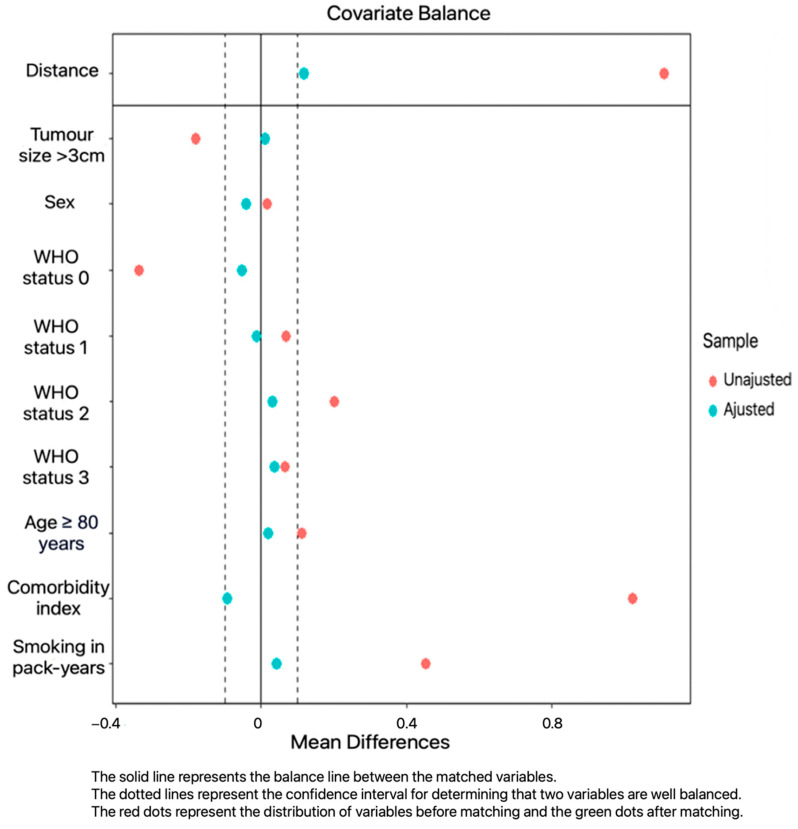
Balance of matching variables.

**Figure 2 cancers-17-00677-f002:**
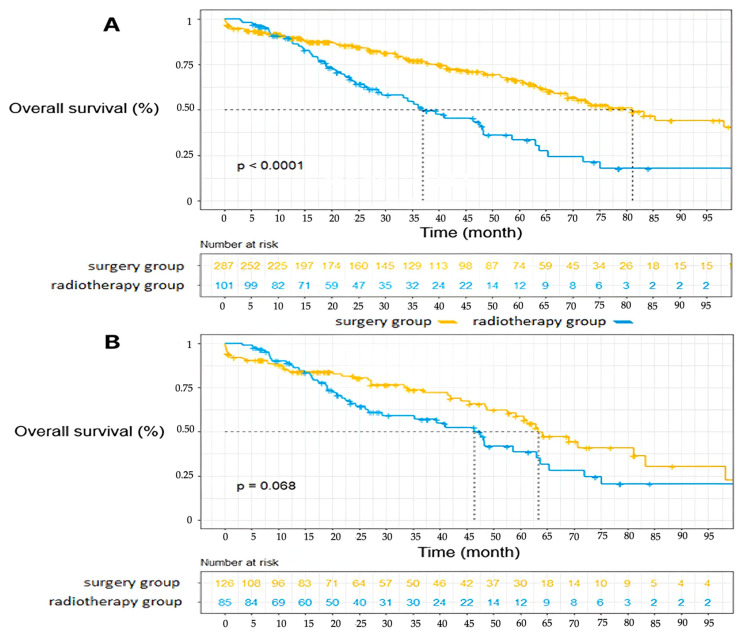
Overall Survival (**A**) Overall survival in the global population (**B**) Overall survival in the matched population.

**Figure 3 cancers-17-00677-f003:**
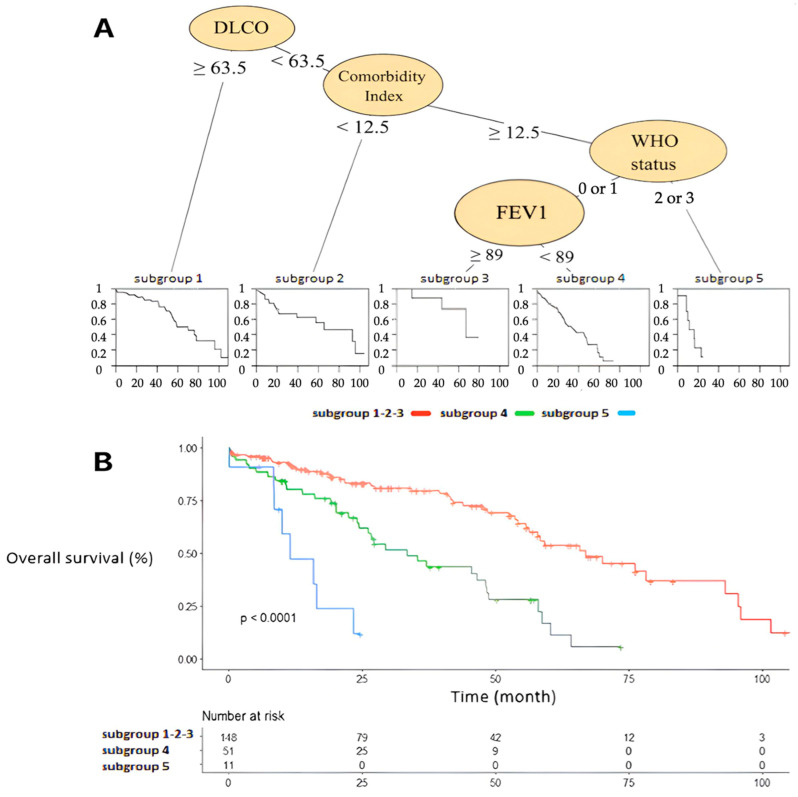
Cluster analysis of survival (**A**) CART analysis of overall survival (**B**) Overall survival curves.

**Figure 4 cancers-17-00677-f004:**
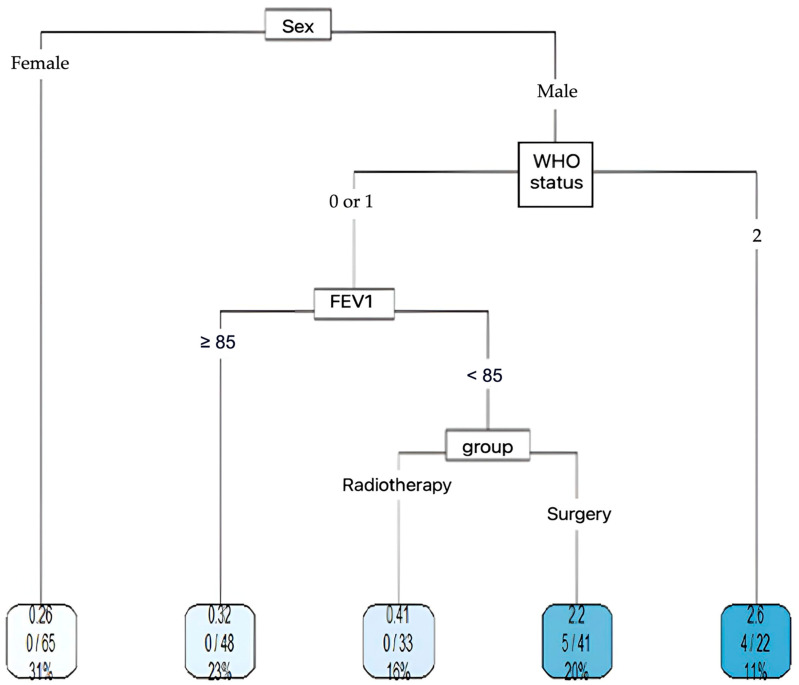
Distribution of early mortality in matched population.

**Table 1 cancers-17-00677-t001:** Characteristics of the overall population pre-matching.

	Surgery Group (n = 288)	Radiotherapy Group (n = 101)	*p* Value
Sex			0.801
Male	188 (65.3%)	68 (67.3%)	
Female	100 (34.7%)	33 (32.7%)	
Age *	78.60	79.44	0.047
Tumor > 3 cm (T2)	92 (31.9%)	14 (13.9%)	0.001
WHO status			<0.001
0	172 (59.7%)	26 (25.7%)	
1	98 (34%)	42 (41.6%)	
2	17 (5.9%)	26 (25.7%)	
3	1 (0.3%)	7 (6.9%)	
Patients without histology	0	51 (50.5%)	
Tumor stage			
IA	196 (68.1%)	87 (86.1%)	
IB	73 (25.3%)	14 (13.9%)	
IIA	19 (6.6%)	0	
Smoking in pack-years *	32.10 (28.50)	44.40 (27.24)	<0.001
BMI *	25.66 (4.0)	25.99 (5.49)	0.542
Comorbidity index *	9.77 (6.90)	15.23 (5.34)	<0.001
DLCO *	70.46 (14.82)	51.41 (16.96)	<0.001
FEV1 *	92.08 (20.80)	77.11 (29.15)	<0.001
FEV/FVC ratio *	74.98 (12.42)	63.74 (13.42)	<0.001

* Average.

**Table 2 cancers-17-00677-t002:** Characteristics of the matched population.

	Surgery Group (n = 127)	Radiotherapy Group (n = 85)	*p* Value
Sex			0.753
Male	89 (70.1%)	57 (67.1%)	
Female	38 (29.9%)	28 (32.9%)	
Âge *	78.86	79.51	0.255
Tumor > 3 cm (T2)	20 (15.7%)	13 (13.3%)	1.000
WHO status			0.081
0	53 (41.7%)	26 (30.6%)	
1	58 (45.7%)	39 (45.9%)	
2	15 (11.8%)	16 (18.8%)	
3	1 (0.8%)	4 (4.7%)	
Patients without histology	0	42 (49.4%)	
Tumor stage			0.229
IA	107 (84.3%)	72 (84.7%)	
IB	16 (12.6%)	13 (15.3%)	
IIA	4 (3.1%)	0	
Smoking in pack-years *	39.65 (29.91)	42.53 (26.77)	0.474
BMI *	26.04 (4.02)	25.42 (5.09)	0.334
Comorbidity index *	14.02 (6.49)	14.48 (4.48)	0.571
DLCO *	67.02 (16.95)	51.45 (17.87)	<0.001
FEV1 *	88.29 (20.35)	78.73 (30.00)	<0.008
FEV/FVC ratio *	74.03 (12.74)	64.21 (13.96)	<0.001

* Average.

**Table 3 cancers-17-00677-t003:** Characteristics of patients treated by surgery or radiotherapy stereotactic ablative radiotherapy.

Factor	Workforce
Surgical approach	
VATS	207 (71.9%)
RATS	44 (15.3%)
Thoracotomy	36 (12.5%)
Sternotomy	1 (0.3%)
Type of resection	
Lobectomy	216 (75%)
Segmentectomy	45 (15.7%)
Wedge resection	23 (8%)
Bilobectomy	3 (1%)
Type of lymph node resection	
Systematic lymphadenectomy	233 (80.9%)
Oriented Lymphadenectomy	34 (11.8%)
Per operative Biopsy	3 (1%)
No lymph node resection	18 (6.3)
Pathological lymph node involvement	
pN1	19 (7.03%)
pN2	16 (5.9%)
cT2 tumors convert to pT1	16 (5.5%)
Quality of resection	
R0	257 (89.2%)
R1	7 (2.4%)
Run	24 (8.4%)
Radiotherapy machine	
Tomotherapy^®^	68 (67.3%)
Cyberknife^®^	20 (19.8%)
VERSA HD^®^	13 (12.9%)
Patients with BED < 100 Gy	22 (21.8%)
Patients with central tumor	28 (27.8%)
Patients with history of radiotherapy	16 (15.8%)
Operable patients	8 (7.9%)

**Table 4 cancers-17-00677-t004:** Multivariate analysis in the matched population for Overall Survival.

**Group**	**Surgery** **n = 127**	**Reference**	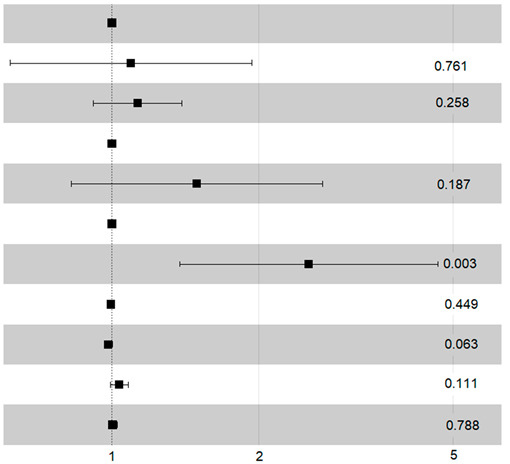
	Radiotherapy n = 85	1.09 (0.62–1.9)	
Tumour Size	n = 212	1.13 (0.92–1.4)	
Sex	Female n = 66	reference	
	Male n = 146	1.49 (0.82–2.7)	
WHO status	0–1 n = 176	reference	
	2–3 n = 36	2.53 (1.38–4.7)	
FEV1	n = 212	1.00 (0.98–1.0)	
DLCO	n = 212	0.98 (0.97–1.0)	
Comorbidity index	n = 212	1.04 (0.99–1.1)	
FEV/FVC	n = 212	1.00 (0.98–1.0)	


## Data Availability

Research data are stored in an institutional repository and will be shared upon request to the corresponding author.

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
