# Peer review of "Surgery Versus Stereotactic Radiotherapy in Patients over 75 Years Treated for Stage IA–IIA NSCLC"

_cancers, 2025, doi:10.3390/cancers17040677_

Round 1
Reviewer 1 Report
Comments and Suggestions for Authors
The manuscript titled "Surgery Versus Stereotactic Radiotherapy in Patients over 75 Treated for Stage IA-IIA NSCLC" can be accepted for publication after some corrections.
1). The introduction section has to be improved with more literature surveys.
2). Figure numbers have been irregularly represented or ordered.
3). The quality of the figures needs to be improved.
4). There are several grammatical and typographical error that needs to be corrected.
Comments on the Quality of English LanguageIt needs to be improved for better understanding
Author Response
Dear reviewer, please find our answers in the attached file.

Reviewer 2 Report
Comments and Suggestions for Authors
This is a large single-center retrospective study on the survival and regional control of patients over 75 years treated 21 by surgery or stereotactic radiotherapy (SABR) for localized NSCLC. Authors demonstrated that surgery seems to remain the standard of treatment in terms of overall survival and locoregional recurrence, in a context where SABR nevertheless provides excellent local control and tolerance in the short and long term. This study may provide some useful information on the treatment modalities for elderly patients diagnosed with localized NSCLC. I have some comments.
<Comments>
1. In the title, please insert “years” behind 75. (75 years)
2. In line 17, Please describe the full term of the abbreviation, “NSCLC”.
3. In line 31, Please describe the full term of the abbreviation, “WHO” and “FEV1”
4. In line 57, please add the “years”. (> 75 year)
5. In line 75, Please describe the full term of the abbreviation, “CT” and “PET-CT”
6. The figures in the text manuscript in line 155 should be cited in consecutive order.
(ex, Fig 1-> 2 -> 3 -> 4)
7. Please describe more details for comorbidity index.
8. In line 180, Please describe the full term of the abbreviation, “VEMS”.
Comments on the Quality of English LanguageThe English could be improved to more clearly express.
Author Response

(The authors gave the same response as above.)

Reviewer 3 Report
Comments and Suggestions for Authors
This is a very well written and illustrated manuscript on an important topic of great interest to clinicians and patients. I have no problem with recommending that it can be published without revision.
I will make a few observations that the authors may wish to consider if revisions are suggested by other reviewers.
This study documents the increased mortality of pulmonary resection in elderly patients with comorbidity and reinforces the survival advantage experience by surgery vs RT. It does not soft pedal the problems of retrospective studies or the necessity of providing RT to patients without preliminary tissue biopsy. I am pleased to note that French thoracic surgeons do a much better job of systematic nodal dissection than in performed in the U.S. Although survival is poorer in the RT group, it is valuable to demonstrate that many elderly LC patients with comorbid disease can be salvaged even when the risk of surgery is high.
In the U.S. there is a marked and growing change as more and more 1 cm. LC detected by screening. Hopefully France will finally adopt population CT screening. The practical impact is that this and other papers do not contain information on survival surg vs RT by size and this is an increasingly important consideration. RT will not cure patients with nodal disease, but N2 is uncommon with 1 cm LC. Can the authors provide any data on survival by tumor diameter in their large series?
Thank you for the privilege of reviewing this most worthy manuscript.
Author Response

(The authors gave the same response as above.)

Reviewer 4 Report
Comments and Suggestions for Authors
I read the manuscript "Surgery Versus Stereotactic Radiotherapy in Patients over 75 treated for Stage IA-IIA NSCLC" by Galdric et al. Overall, this is an interesting study. Below, I leave some minor comments so the authors can consider them:
- I suggest authors add additional keywords such as "real-world evidence," "lung surgery," "propensity score matching," "tree-based model," and "French."
- In the statistical analysis section, please report how the authors manage missing data due to the study's retrospective nature.
- Please change the labels of Figure 4 (Tailler T_rec_<3, sexe, OMS_0, etc ....) by the parameter reported in the text (tumour size >3, sex, WHO status 1 and so on) or report a legend in the figure.
- In the analysis, surgical patients and patients treated with radiotherapy were matched 2:1 by propensity score for variables of interest, which were evenly distributed in the pre-matching population. Conversely, unbalanced variables such as DLCO, VEMS and FEV/FVC ratio were not included as regressors in the propensity score match. Please explain this choice and how these unbalanced variables could impact the results. Whether these variables were included in the univariate and multivariate Cox analysis is unclear.
- In the Table 1 caption, please detail the" Characteristics of the overall population pre-matching."
- In Tables 1 and 2, replace the word gender with sex.
- In Tables 1 and 2, please add standard deviation when appropriate.
- In Table 2, please replace Moyennes with Average.
- In the results section, in addition to the median follow-up, please add the minimum and maximum range for both groups.
- Please check the correctness of the resection quality parameters' names in Table 3 and add the % symbol where it is missing.
- In the whole manuscript, the authors never reported the confidence intervals CI. Please report CI when appropriate.
- I suggest displaying the Cox analysis results through a forest plot because it is unclear which risk factors were included in the univariate and multivariate Cox analyses for OS and locoregional recurrences. The results section poorly describes the multivariate and univariate results.
Author Response

(The authors gave the same response as above.)
